Effects of 12-week integrative neuromuscular training on muscular fitness and sex differences in response to intervention in five- to six-year-old preschoolers

Wang Zhihai 1
Zang Jiayu 1
Wang Zhaohong 2
Fong Daniel T.P. 3
Wang Dan wangdan@sus.edu.cn 1
1 School of Athletic Performance, Shanghai University of Sport , Shanghai , China
2 Fifth Experimental Kindergarten, Xin’an County, Luoyang City , Henan , China
3 National Centre for Sport and Exercise Medicine, School of Sport, Exercise and Health Sciences, Loughborough University , Loughborough , United Kingdom
Espada Mário
Electronic publication date: 2025 May 8
Publication date: 2025
Volume: 13
Electronic Location ID: e19417
Received 2024 Dec 19; Accepted 2025 Apr 10
Copyright: ©2025 Wang et al.
Copyright year: 2025
Copyright holder: Wang et al.
License: This is an open access article distributed under the terms of the Creative Commons Attribution License, which permits unrestricted use, distribution, reproduction and adaptation in any medium and for any purpose provided that it is properly attributed. For attribution, the original author(s), title, publication source (PeerJ) and either DOI or URL of the article must be cited.
License URL: https://creativecommons.org/licenses/by/4.0/

Keywords: Children, Motor development, Strength training, Neuromuscular exercise, Physical education

Funding: The Program for Overseas High-level Talents at Shanghai Institutions of Higher Learning TP2019072 The Research and Innovation Grant for Graduate Students, Shanghai University of Sport YJSCX-2024-026 The Shanghai Key Lab of Human Performance (Shanghai University of Sport) 11DZ2261100 This work was supported by the Program for Overseas High-level Talents at Shanghai Institutions of Higher Learning under Grant No. TP2019072; the Research and Innovation Grant for Graduate Students, Shanghai University of Sport (Project No. YJSCX-2024-026); and the Shanghai Key Lab of Human Performance (Shanghai University of Sport) under Grant No. 11DZ2261100. The funders had no role in study design, data collection and analysis, decision to publish, or preparation of the manuscript.

==============================
Objectives

This study examined the effects of a 12-week integrative neuromuscular training (INT) program on muscular fitness in male and female five- to six-year-old preschoolers.

Methods

Thirty preschoolers were randomly assigned to either an experimental group (EG; n = 15; age = 5.3 ± 0.5 years, body height = 115.3 ± 5.2 cm, body mass = 20.7 ± 2.6 kg) or a control group (CG; n = 15; age = 5.2 ± 0.4 years, body height = 118.5 ± 4.9 cm, body mass = 22.6 ± 2.7 kg) participating in a 12-week INT program and regular physical education classes three times per week, respectively. Upper extremity maximal strength (grip strength test) and power (tennis ball throwing test), core endurance strength (one-minute sit-up test), and lower extremity power (standing long jump test) were assessed at the baseline (T0), Week 6 (T6), and Week 12 (T12). Data were analyzed using an independent samples T-test and a two-way repeated-measures ANOVA.

Results

Significant interaction effects between the EG and CG were observed for grip strength, tennis ball throws, one-minute sit-ups, and standing long jumps (p < 0.001). Relative to the CG, the EG demonstrated significant improvements in all muscular fitness at T6 and T12 (p < 0.05). However, no significant interaction was found between the time and the sex (p > 0.05).

Conclusions

These findings suggested that a 12-week INT program can more effectively enhance the muscular fitness of 5–6-year-old preschoolers compared to regular physical education classes, serving as an effective and efficient supplement to physical education for this age group. Furthermore, there is no evidence of sex -specific differences in the development of muscular fitness among 5–6-year-old preschoolers under the INT program.

Introduction

Preschool is a period of rapid growth and development, as well as a crucial time for establishing healthy lifestyle behaviors (García-Hermoso et al., 2020; Zou et al., 2024). However, recent studies indicate a concerning decline in physical fitness—particularly in muscular fitness, which includes muscular strength, muscular power, and local muscular endurance (Stricker, Faigenbaum & McCambridge, 2020) among preschool children, which remains unaddressed (Hacke et al., 2019; Parrish et al., 2022; Zhang et al., 2022). For preschool children aged five to six years, this stage represents a crucial period in their transition from early childhood to primary school (Zhou et al., 2016).

As a vital component of overall physical health, muscular fitness plays a determinative role in the development of physical performance (Jiménez-Pavón et al., 2012). Moderate improvements in muscular fitness during this critical growth phase have been strongly linked to healthy bone development, improved muscle coordination, and a reduced risk of sports-related injuries (Cohen et al., 2011; Faigenbaum & Myer, 2010). Conversely, poor muscular fitness during childhood increases the risk of cardiovascular disease and overall mortality in adulthood, imposing substantial burdens on families and society (Artero et al., 2011; García-Hermoso et al., 2018; Ortega et al., 2012).

Previous research has shown that, under professional guidance, regular engagement in traditional resistance training can safely and effectively enhance muscular fitness among school-aged children and adolescents (Faigenbaum & Myer, 2010; Faigenbaum et al., 1999). However, due to the heightened neural plasticity observed in preschool-aged children, it is recommended to employ more targeted interventions for this specific population (Myer et al., 2011b). Integrated neuromuscular training (INT) differs from traditional strength and conditioning training and physical education classes by offering a structured approach to augment muscular fitness and motor control, thereby optimizing them at this critical developmental window (Myer et al., 2011b). INT differs from traditional strength and conditioning training and physical education classes. It integrates elements of strength and conditioning training aimed at developing strength and power with the motor competence components of physical education, creating a synergistic effect that enhances both physical and motor development (Duncan, Hames & Eyre, 2019). Therefore, participating in INT during childhood and adolescence can yield long-term benefits, promoting healthy outcomes in adulthood (Myer et al., 2011b).

Conversely, a lack of pre-adolescent INT could increase the risk of musculoskeletal injuries in youths (Faigenbaum et al., 2014; Hewett et al., 2005). As the primary venues for children’s interactions outside the family, schools provide physical education classes as the most effective way to improve young individuals’ physical fitness (Xiong et al., 2022). Currently, preschool physical education largely consists of unstructured free play, which may hinder the development of children’s muscular fitness due to the lack of regular, organized INT (Gu & Xie, 2024; Määttä et al., 2018). Studies have shown that incorporating INT into physical education curricula significantly enhances motor skills and physical performance in seven- to eight-year-old children (Faigenbaum et al., 2011; Faigenbaum et al., 2014; Sinđić et al., 2021).

INT integrates general (e.g., fundamental motor skills) and specific physical training activities (e.g., resistance, dynamic stability, core strength, plyometrics, and agility; Myer et al., 2015; Wang et al., 2022) and also has been validated to be an advanced and effective method to enhance muscular fitness in children and adolescents (Duncan, Hames & Eyre, 2019; Faigenbaum et al., 2011; Vasileva et al., 2024). This comprehensive approach aims to enhance physical health and promote health-related motor skills (Myer et al., 2015; Wang et al., 2022). Studies suggest that initiating INT before puberty significantly benefits children’s physical health (Myer et al., 2011b). Compared to traditional physical training and physical education classes, INT fosters the concurrent development of strength and motor skills, resulting in synergistic improvements in physical and motor competencies (Duncan, Hames & Eyre, 2019). Moreover, INT is recognized for its safe, cost-effective, flexible practice locations (Duncan, Eyre & Oxford, 2018; Lin et al., 2022)„ and therefore it effectively addresses challenges in physical education classes, such as limited resources and time constraints (Duncan, Eyre & Oxford, 2018; Lin et al., 2022). Notably, the impact of INT may also vary by sex. This phenomenon may be attributed to the fact that, during early childhood, girls often develop fine motor skills and coordination at a younger age, which could facilitate more pronounced neuromuscular adaptations in response to INT (Faber et al., 2024). Faigenbaum et al. (2014) observed that seven-year-old girls who participated in an eight-week INT program twice weekly showed greater improvements in core strength and aerobic capacity than their male counterparts. However, research on preschool children is very limited, creating a gap in our understanding of whether similar trends are present in this younger population. This underrepresentation may be attributed to historical concerns that resistance training could be unsafe for children and potentially harmful to developing musculoskeletal systems (Faigenbaum & Myer, 2010).

Therefore, this study aimed to examine the effects of INT on muscular fitness and explore potential sex differences in preschoolers aged five to six years. We hypothesized that INT enhances muscular fitness in both sexes, with girls showing greater sensitivity to the INT stimulus than boys.

Materials & Methods

Participants

Thirty children (16 girls and 14 boys) were recruited from public kindergartens in Henan Province, China (see Table 1). A sample size calculation (G*Power 3.1.9.2; Franz-Faul, Universität Kiel, Germany, α = 0.05, power = 0.80, and effect size of 0.50 (Duncan, Hames & Eyre, 2019) indicated that a minimum of 10 participants was needed for our study. The participants were randomly allocated into either an experimental group (EG; seven boys, eight girls) or a control group (CG; seven boys, eight girls) using computer-generated random numbers. The EG underwent INT three times weekly, while the CG participated in the regular physical education classes provided by the school. Figure 1 illustrates the enrollment, allocation, and attrition of participants. The inclusion criteria were as follows (Vasileva et al., 2024): (1) aged between five to six years; (2) having no injuries in the past three months; and (3) having no developmental disabilities. Exclusion criteria included the following (Vasileva et al., 2024): (1) major congenital malformations or severe cognitive deficits; (2) musculoskeletal or neurological disorders or long-term medication usage; (3) prior participation in organized INT within three months before the study; (4) any contraindications to exercise; or (5) lack of informed consent from parents or guardians. This study was approved by the Human Research Ethics Committee of Shanghai University of Sport (No. 102772023RT008). Written consent was secured from all parents/guardians and children before commencing the study.

Table 1 Demographic information of the participants.

Variables	EG (n = 15)	CG (n = 15)	p -value	
Age (y)	5.3 ± 0.5	5.2 ± 0.4	0.679	
Height (cm)	115.3 ± 5.2	118.5 ± 4.9	0.10	
Body mass (kg)	20.7 ± 2.6	22.6 ± 2.7	0.161	
BMI (kg/m2)	15.5 ± 1.8	16.0 ± 1.5	0.415	
Notes.

Values are reported as means ± standard deviations (SDs).

EG integrated neuromuscular training

CG regular physical education class

BMI body mass index

Figure 1 The flowchart of participant recruitment, randomization, and flow during the study.

Study design

This is a 12-week single-blind randomized controlled trial. Researchers were aware of group allocations; however, participants were blinded to the potential benefits of the interventions or the study’s hypotheses. Assessments were conducted at three-time points—the baseline (T0), Week 6 (T6), and Week 12 (T12)—in both the experimental group (EG) and the control group (CG). The evaluations included anthropometric measurements (height and weight), muscular fitness assessments, including upper extremity strength and power (grip strength and tennis ball throw), core endurance strength (one-minute sit-up test (Mačak et al., 2022)), and lower extremity power (standing long jump). The test was selected according to the evaluation methods specified in China’s National Physical Fitness Testing Standard (Children’s Part), which is commonly used to assess the level of children’s muscular fitness (Duncan, Hames & Eyre, 2019; Lv et al., 2023; Mačak et al., 2022). Before the intervention, the participants attended familiarization sessions to practice and fully understand the protocol. The test began with a 10-minute dynamic warm-up, which included five minutes of jogging and five minutes of whole-body dynamic activation exercises (e.g., lunges, butt kicks, jumping jacks, etc.). Once the participants had warmed up and familiarized themselves with the testing movements, they completed the test series sequentially (see Fig. 2). To maintain consistent testing conditions, each session was performed at the same time of day under identical environmental conditions. The testing process was consistently supervised and recorded by the same researcher to ensure data accuracy and reliability. The participants and their guardians were instructed to adhere to a consistent dietary program and refrain from participating in additional physical activities beyond the regular physical education curriculum. Compliance was monitored through verbal reminders and follow-up phone calls every week.

Figure 2 The flowchart of the test.

Anthropometric assessments

Height and weight measurements were obtained using an electronic height and weight meter (Kaiyuan Electronics HW-600; Kaiyuan Electronics, Zhengzhou, China). Participants were assessed barefoot and dressed in lightweight, sport-appropriate attire. The device provided measurement accuracies of 0.1 kg for weight and 0.1 cm for height. Body mass index was calculated using the formula: weight (kg)/(height (m))2 (Duncan, Stanley & Wright, 2013).

Upper extremity strength assessments

Upper extremity strength tests consisted of an upper extremity maximal strength test (grip strength test) and a power test (tennis ball throwing test). The grip strength test was conducted using a grip strength dynamometer (Xinman WCS-100, Shanghai Xinman Science and Education Equipment Co., Ltd., China) (Mačak et al., 2022). Participants were guided to look forward in a standing position with their feet shoulder-width apart and extend the elbow on the side of their dominant hand. The dominant hand was discerned by which hand the participant used to throw a ball (Valldecabres, Richards & De Benito, 2022). They were then directed to exert maximum force by squeezing the dynamometer’s handle. Each participant performed this test twice, with an interval of one minute between attempts to minimize fatigue. The peak value of these two attempts was recorded for further analysis, with a precision of 0.1 kg. For the evaluation of upper extremity explosive strength, the tennis ball throwing test was used (Li et al., 2024). In this test, the participants stood behind a marked throwing line, faced the intended direction of the throw, and adopted a split stance. Holding a tennis ball in their dominant hand, they threw it backhand, aiming for the farthest distance. The best result from the two attempts, measured accurately to 0.1 m, was recorded as the final measure.

Lower extremity power assessments

The standing long jump test was used to assess lower extremity power in children (Li et al., 2024). Before the test, participants were instructed to stand with their feet parallel and shoulder width apart. During the jump, they were required to take off and land with both feet simultaneously. Participants were allowed to swing their arms to gain momentum during the jump. The jump distance was measured from the tips of the toes at takeoff to the heels at landing. If the feet did not land in parallel, the heel of the rearward foot was used for measurement. This measurement was taken using a tape measure (Deli 8203, Deli Group Co., Ltd., China) with an accuracy of 0.01 m. If a participant’s hands touched the ground during the landing phase, the attempt was deemed invalid. Each child performed two attempts, with a three-minute rest between trials, and the best result was recorded for the final analysis.

Core endurance strength assessments

The one-minute sit-up test was used to assess core endurance strength in children (Oliver & Di Brezzo, 2009). Participants were positioned lying on a mat with their knees bent at right angles, feet flat on the floor, and held by a partner to ensure proper form and stability during the test. A sit-up was deemed completed when the participants made contact between their elbows and knees, followed by the return of their shoulders to the ground. To motivate the participants to maximize their efforts, verbal encouragement was provided by the researcher, who also announced the remaining time every 30 s. Each participant underwent the test once, with the total number of valid sit-ups executed in 60 s recorded using a stopwatch (model PC100B, manufactured by Fuhai Chemical Glass Instrument Co., Ltd., China) for subsequent statistical analysis.

Training program

The INT program in this study was specifically designed for preschool children based on early childhood resistance training research and the physical and mental development characteristics of five- to six-year-olds (Duncan, Eyre & Oxford, 2018; Faigenbaum et al., 2011; Faigenbaum et al., 2014). It focused on bodyweight exercises for strength development while integrating exercises that enhance balance, agility, and coordination. To make training more engaging and to improve participants’ comprehension of movements, researchers assigned imaginative names and scenarios for each exercise. The program was led by certified physical education teachers and experienced youth strength and conditioning coaches, ensuring an instructor-to-trainee ratio of 1:8. Sessions were held three times per week—on Mondays, Wednesdays, and Fridays—each lasting 40 min over 12 weeks. This schedule served as an alternative to the standard school physical education curriculum, ensuring that no conflicts arose. Training sessions were spaced 48 h apart to optimize participants’ recovery. During the 12-week intervention, participants missing more than two sessions were excluded from the final analysis.

The INT program began with a 10-minute dynamic warm-up consisting of jogging and dynamic stretching exercises. This was followed by a specialized 25-minute INT program that targeted various muscle groups—upper extremity, lower extremity, and the core—on alternating days (Mondays, Wednesdays, and Fridays), focusing on strengthening the respective muscle groups. To amplify neuromuscular stimulation, the INT program gradually escalated in both training volume (e.g., repetitions) and intensity (e.g., difficulty of exercises and decreased interval duration) across three phases: an adaptation and familiarization phase in weeks 1–4, a progression phase in weeks 5–8, and a strengthening phase in weeks 9–12. For instance, the number of sets performed by participants increased from two to three over 12 weeks, while rest intervals diminished from one minute to 20 s. A detailed description of the INT program’s overall structure is provided in Table 2. Examples of exercises are illustrated in Fig. 3.

Table 2 Structure of the integrative neuromuscular training program.

Phase	Description	
	Monday (upper extremity training)	Wednesday (lower extremity training)	Friday (core training)	
Adaptation and familiarization phase (weeks 1–4)	Incline push-ups (superhero upper body push) ∗ six reps	Frog jumping ∗ 6 m	Supine cycling (monkey cycling) ∗ 6 s	
Donkey kick (donkey jump) ∗ six reps	Straight-line bipedal hopping (rocket bunny jumping) ∗ 6 m	Crunches (Panda wakes up) ∗ six reps	
Crawling (Bear goes to school) ∗ 6 m	Vertical jump (monkey picking peaches) ∗ six reps	Prone jumping jack (little tailor) ∗ six reps	
				
Progression phase (weeks 5–8)	Push-ups with alternating shoulder taps (Superman’s transformation) * eight reps	Shuttle run (little speedster) ∗ 8 m	Plank tuck jump (little groundhog) ∗ eight reps	
Supine four-point brace walk (turtle crawling) ∗ eight reps	Straight-line single-leg hopping (kangaroo jumping) ∗ 8 m	Mountain running (bear climbing) ∗ 8 s	
Lateral crawling (crab crawling) ∗ 8 m	Jump box (magic jumping box) ∗ eight reps	Supine leg raises (dolphin-style water kicking) ∗ eight reps	
				
Strengthening phase (weeks 9–12)	Push-ups with alternating shoulder taps (Superman’s transformation) ∗ 12 reps	Shuttle run (little speedster) ∗ 12 m	Plank tuck jump (little groundhog) ∗ 12 reps	
Supine four-point brace walk (turtle crawling) ∗ 12 reps	Single-leg lateral jump (bunny hop) ∗ 12 m	Mountain running (bear climbing) ∗ 12 s	
Lateral crawling (crab crawling) ∗ 12 m	Jump box (magic jumping box) ∗ 12 reps	Supine leg raises (dolphin-style water kicking) ∗ 12 reps	
Notes.

Reps repetitions

Figure 3 Examples of integrative neuromuscular training.

(A) Push-ups with alternating shoulder taps; (B) crawling; (C) supine four-point brace walk; (D) donkey kick; (E) vertical jump; (F) frog jumping; (G) supine leg raises; and (H) mountain running.

The CG participated in three 40-minute regular physical education classes per week. Each class included a 10-minute dynamic warm-up, 25 min of free play or elementary games using kindergarten equipment (e.g., slides, sandpits, swings, balls, bricks, and toys), and a 5-minute cool-down with stretching. The free play consisted of traditional, fun, and engaging activities such as relay games, mirror games, change places, hopscotch, and musical chairs.

Considering the young age of our participants, the Borg rate of perceived exertion (Borg CR-10; rating of perceived exertion (RPE); (Borg, 1982)) and the Facil RPE (Huang, Chiou Joi & engineering p, 2013) were utilized to monitor the intensity of exercises. The intensity was maintained at medium to high levels (i.e., RPE of 3–6 and Facil RPE score of 6).

Statistical analysis

Data were analyzed using SPSS 26.0 (SPSS Inc., Chicago, IL, USA). Descriptive statistics (means and standard deviations) were used to summarize the data. Normality was assessed using the Shapiro–Wilk test, and homogeneity of variance was tested using Levene’s test. If the assumption of normality was violated, the Kruskal-Wallis test was applied. Baseline differences between the EG and CG were analyzed using an independent samples t-test. A two-way repeated measures ANOVA (3-time points (T0, T6, T12) × 2 groups (EG vs. CG)) was employed to assess the effect of INT on muscular fitness. Additionally, to further investigate the effect of INT on the muscular fitness of preschool children by sex, a two-way repeated measures ANOVA (3-time points (T0, T6, T12) × 2 sexes (boys vs. girls)) was conducted again. Greenhouse-Geisser corrections were applied if sphericity assumptions were violated. If a significant interaction effect between time and group was observed, post-hoc pairwise comparisons (Bonferroni-adjusted) were performed to examine within- and between-group differences. Effect sizes were calculated using partial eta squared (η2p), with values interpreted as small (0.01), moderate (0.06), and large (0.14; (Cohen, 2013)). The significance level was set at α = 95%, and p < 0.05 was considered statistically significant.

Results

All 30 participants completed the 12-week training program without any reported adverse events, and the compliance rate was 100% since the study was conducted within the kindergarten semester.

No significant differences were observed between the EG and the CG in any variables analyzed pre-intervention. Significant interactions between the time and the group were observed in the following: grip strength (F = 15.320, p < 0.001, η2p = 0.354); tennis ball throw (F = 69.907, p < 0.001, η2p = 0.400); one-minute sit-up (F = 18.698, p < 0.001, η2p = 0.714); and standing long jump (F = 33.104, p < 0.001, η2p = 0.542; see Fig. 4). Post-hoc tests revealed that the EG showed greater improvements in grip strength (T0 vs. T6: p = 0.002, T6 vs. T12: p = 0.004), tennis ball throw (T0 vs. T6: p < 0.001, T6 vs. T12: p < 0.001), one-minute sit-up (T0 vs. T6: p < 0.001, T6 vs. T12: p < 0.001), and standing long jump (T0 vs. T6: p < 0.001, T6 vs. T12: p < 0.001) at T6 and T12 compared to the CG. In the EG, compared to T0, grip strength showed a significant increase at T6 (p = 0.002) and T12 (p < 0.001), with improvements of 0.8 kg (95% CI [0.27–1.27]) and 1.9 kg (95% CI [1.19–2.57]), respectively. Between T6 and T12, the grip strength further increased significantly (p < 0.001) by 1.1 kg (95% CI [0.70–1.53]). In terms of the tennis ball throwing distance, significant enhancements were observed at both T6 (p < 0.001) and T12 (p < 0.001), with respective increases of 1.6 m (95% CI [1.15–1.99]) and 2.7 m (95% CI [2.26–3.12]). From T6 to T12, there was a significant improvement (p < 0.001) of 1.1 m (95% CI [0.88–1.36]) in the throwing distance. For one-minute sit-ups, significant improvements were found at T6 (p < 0.001) and T12 (p < 0.001), with increases of 5.9 (95% CI [4.39–7.47]) and 9.9 repetitions (95% CI [7.00–12.86]), respectively. The number of one-minute sit-ups performed further increased significantly (p = 0.002) by four repetitions (95% CI [1.38–6.62]) from T6 to T12. Regarding the standing long jump, significant improvements were noted at T6 (p < 0.001) and T12 (p < 0.001), with respective 0.22 m (95% CI [0.15–0.28]) and 0.31 m (95% CI [0.23–0.40]). Between T6 and T12, the standing long jump distance increased significantly (p < 0.001) by 0.10 m (95% CI [0.06–0.13]).

Figure 4 (A–D) Time × group interaction effects on muscle strength indicators for both the EG and CG at baseline, after 6 weeks of intervention, and after 12 weeks of intervention.

T0, pre-intervention; T6, six weeks post-intervention; and T12, 12 weeks post-intervention. * Significant interaction between the time and the group.

No significant interaction existed between the time and sex were observed in the grip strength (p = 0.823), tennis ball throw (p = 0.366), one-minute sit-up (p = 0.728), and standing long jump (p = 0.315), indicating that the response to training was similar between boys and girls in the INT group. The main effects of time on grip strength (F = 37.55, p < 0.001, η2p = 0.743); tennis ball throw (F = 97.874, p < 0.001, η2p = 0.883); one-minute sit-up (F = 35.46, p < 0.001, η2p = 0.732); and standing long jump (F = 50.745, p < 0.001, η2p = 0.796) results were all significant. There were no significant main effects of sex on grip strength, tennis ball throw, one-minute sit-up, and standing long jump results (p > 0.05). After 12 weeks of INT, boys showed greater improvements in grip strength (boys: 7.2 to 9.1 kg (95% CI [0.94–3.00]), girls: 5.5 to 7.3 kg (95% CI [0.84–2.76)) and one-minute sit-ups (boys: 2.3 to 12.6 repetitions (95% CI [4.43–16.15]); girls: 7.1 to 16.8 repetitions (95% CI [4.14–15.11])) than girls. Girls showed greater improvement than boys in tennis ball throws (girls: 3.3 to 5.8 m (95% CI [1.69–3.36]), boys: 4.3 to 8.2 m (95% CI [1.98–3.77])) and standing long jumps (girls: 0.9 to 1.3 m (95% CI [0.20–0.51]), boys: 0.9 to 1.2 m (95% CI [0.11–0.43])).

Discussion

The purpose of this study was to determine the effects of 12 weeks of INT on the muscular fitness of preschool children. Our findings showed that a 12-week INT program effectively enhances the upper and lower extremity, as well as core muscle strength, in five- to six-year-old preschoolers compared to a regular school physical education program. Notably, significant improvements in muscular fitness were observed after just six weeks of INT. This suggests that even a relatively short intervention period can yield measurable benefits. Moreover, there was no significant difference in the sensitivity to INT between preschool girls and boys, suggesting that both sexes exhibited high adaptability to INT interventions and thus demonstrated comparable levels of neuromuscular plasticity. To the best of our knowledge, this is the first study aiming to explore the effects of INT on muscular fitness in five- to six-year-old preschoolers. In general, these findings confirmed our hypothesis that INT is a safe, efficient, and cost-effective way to promote muscular fitness improvement in preschoolers under the guidance of a professional.

In contrast to previous investigations examining INT’s effects on physical health and health-related motor skills in children and adolescents, the present study specifically elucidates INT’s efficacy in enhancing muscular fitness among a younger demographic (preschool children aged 5–6 years). To our knowledge, this investigation yields three seminal contributions to the literature: (a) it provides the first evidence establishing INT’s effectiveness for preschool-aged populations; (b) reveals the sex-independent nature of training responsiveness, indicating comparable neuromuscular plasticity between sexes; and (c) offers empirically validated insights for developing evidence-based muscular fitness protocols tailored to early childhood development. Furthermore, these findings offer critical empirical support for implementing INT as a scientifically validated supplement to regular preschool physical education curricula, with practical applications for warm-up activities, recess periods, and after-school programming.

This study’s findings suggested that INT conducted three times a week for 12 weeks is more effective than regular physical education classes in enhancing upper and lower extremity and core muscle strength in preschoolers. This indicates that preschool children display a specific response to INT that enhances their muscular fitness, including both muscle strength and power. The results of this study are consistent with those reported by Duncan, Eyre & Oxford (2018) who found that a 10-week INT program significantly improved six- to seven-year-old children’s performance in the medicine ball throw, standing long jump, and countermovement tests compared to regular physical education classes. Muscle strength is primarily influenced by neural and morphological factors, with the development of neural variables being a key factor in improving children’s strength performance (Ramsay et al., 1990).

The pre-adolescent period is a sensitive phase for a child’s physical fitness development and motor skill learning, laying the foundation for health in adolescence and beyond. During this stage, the brain’s neural plasticity provides a window of opportunity to develop muscular fitness (Raudsepp & Jürimäe Bos, 1996). Brain structures or functions are particularly sensitive to specific external stimuli, facilitating the establishment of neuromotor circuits (Raudsepp & Jürimäe Bos, 1996). The scientific design of intervention programs may be crucial for this success. Consequently, many factors should be considered when designing and advancing a program, such as the developmental and fitness levels of participants, choice of exercise, and progression of training loads (Myer et al., 2011a). The INT program in this study was designed as an organized, progressive training approach based on the physical and mental developmental characteristics and strength developmental patterns of five- to six-year-old children. Integrating resistance, power, agility, balance, and speed training into this progressively INT program significantly enhances participants’ neuromuscular control and postural control capabilities. Furthermore, our training protocol is both challenging and enjoyable, with each exercise contextualized within a specific scenario to sustain children’s engagement and motivation. INT can effectively enhance the neuromuscular control abilities of preschool children (aged over seven years old; Ozmun, Mikesky & Surburg, 1994) and the activation, coordination, recruitment, and discharge abilities of muscle motor units (Ramsay et al., 1990). Additionally, improvements in postural control may influence the enhancement of muscular fitness (Hamed et al., 2018). Several studies have found that school-based INT notably enhances postural control abilities and increases muscular fitness in children aged eight years old (Guzmán-Muñoz et al., 2020; Sinđić et al., 2021).

Interestingly, the INT program demonstrated markedly greater improvements in children’s muscular fitness than regular physical education classes after only six weeks. In general, the findings of this study highlighted the efficacy and time-efficient advantages of INT in enhancing muscular fitness among five- to six-year-old preschool children. These results corroborate and extend previous studies that indicate that an eight- to 10-week period of INT, when integrated into or replacing physical education class, can induce positive changes in children (Faigenbaum et al., 2011; Faigenbaum et al., 2014; Sinđić et al., 2021). Faigenbaum et al. (2014) observed that an eight-week INT program lasting 15 min before physical education class improved core and lower extremity strength in seven- to eight-year-old preschool children. However, for five- to six-year-old preschoolers, just six weeks of INT were sufficient to effectively enhance their muscular fitness. This age group represents a sensitive and critical period for children’s development and motor skill acquisition, with the brain’s neuroplasticity offering a window of opportunity for the development and reinforcement of muscular fitness (Ramsay et al., 1990; Raudsepp & Jürimäe Bos, 1996). Improvements in strength parameters among preschool children are primarily due to short-term enhancements in neuromuscular control (Ozmun, Mikesky & Surburg, 1994). Furthermore, substituting part of the physical education session with INT may yield greater benefits in improving muscular fitness than conducting INT during the initial 15 min of physical education class.

This study revealed that, despite the absence of a statistically significant difference in the benefits of INT between boys and girls, after 12 weeks of INT, boys demonstrated greater improvement in grip strength (27.5% vs. 24.8%) and one-minute sit-ups (448.9% vs. 134.9%) than girls. Conversely, girls demonstrated greater improvement than boys in tennis ball throws (77.3% vs. 66.1%) and standing long jumps (36.1% vs. 29%). Although boys and girls showed slightly different patterns of improvement, no significant time × sex interaction was found. This suggests that both sexes responded similarly to INT, despite some variations in absolute gains. Our findings support and extend the observations of Faigenbaum et al. (2014), who noted that seven-year-old girls surpassed boys in gains of lower extremity power following an eight-week combined INT and physical education program (INT was conducted during the first 15 min of each 43-minute regular physical education class; (Faigenbaum et al., 2014). They suggested that this age group of girls might be experiencing a brief, sex-specific developmental phase, rendering them more sensitive to INT interventions (Faigenbaum et al., 2014). Our research further infers that this specific developmental period might also be present in five- to six-year-old preschool girls. However, variations in the age of participants and intervention protocols hinder direct comparisons with Faigenbaum et al.’s (2014) study. Consequently, future longitudinal studies are needed to further explore the sex-specific role of INT in preschoolers.

There were several limitations to the present study that should be acknowledged. First, the small sample size may increase the risk of Type II errors, particularly in the analysis of sex differences. Future studies should consider employing larger sample sizes to enhance the reliability and generalizability of the findings. Second, both the experimenters and testers were aware of group allocations, which may have introduced potential bias in the assessment process. Third, this study primarily focused on the influence of INT on muscular fitness in preschool children, leaving other aspects of physical fitness, such as endurance and agility, unexplored. Expanding the scope of future investigations to include these dimensions would provide a more comprehensive understanding of INT’s effects. Finally, the absence of follow-up sessions after the 12-week intervention period limits our ability to determine whether the observed positive changes are sustained over time. Longitudinal studies with extended follow-up periods are recommended to assess the long-term benefits of INT.

Conclusions

Our findings indicate that INT is more effective than regular physical education in improving upper extremity, lower extremity, and core muscle strength performance in preschool children aged five to six years and is a safe, efficient, and cost-effective method of enhancing their muscular fitness. Notably, there was no significant difference in the sensitivity of five- to six-year-old girls and boys to the effects of INT performed three times weekly. These findings will help physical education teachers and strength and conditioning coaches design effective fitness programs for preschoolers and optimize muscular fitness training adaptations for preschoolers. Future research should further investigate the long-term effects of INT on strength and other health-related effects in preschoolers.

Practical applications

The findings of this study suggest that a thrice-weekly, 12-week INT program significantly improves the muscular fitness of five- to six-year-old preschool children compared to regular physical education classes. Notably, similar enhancements can be achieved with a six-week INT program. These results support the effectiveness and feasibility of INT as a safe and efficient exercise approach, making it an ideal supplement to kindergarten physical education curricula. Moreover, these findings will assist physical education teachers and fitness trainers in designing effective fitness programs for young children and enhancing training adaptability during early childhood, which may have potential short-term and long-term health benefits for preschool children. While INT cannot fully replace traditional physical education classes, it can be effectively integrated into school activities, such as warm-ups before physical education classes, recess periods, or after-school programs.

Physical education teachers and youth strength and conditioning coaches should design, implement, and integrate safe, age-appropriate, effective, and enjoyable programs that provide measurable health and fitness benefits while minimizing injury risks associated with unsafe exercise environments, poor technique, excessive load or volume, and inadequate supervision. We recommend that kindergartens establish dedicated training areas free from environmental hazards, with sufficient space to accommodate all participants (e.g., large classrooms). Furthermore, regular professional training for physical education teachers and strength and conditioning coaches will help ensure the safety and effectiveness of the training programs.

Supplemental Information

Supplemental Information 1 Raw measurements

We thank all the teachers, students, and researchers for their voluntary participation in this study.

Additional Information and Declarations

Competing Interests

Author Contributions

Human Ethics

Data Availability

The authors declare there are no competing interests.

Zhihai Wang conceived and designed the experiments, performed the experiments, analyzed the data, prepared figures and/or tables, authored or reviewed drafts of the article, and approved the final draft.

Jiayu Zang conceived and designed the experiments, performed the experiments, prepared figures and/or tables, and approved the final draft.

Zhaohong Wang conceived and designed the experiments, prepared figures and/or tables, and approved the final draft.

Daniel T.P. Fong analyzed the data, authored or reviewed drafts of the article, and approved the final draft.

Dan Wang conceived and designed the experiments, performed the experiments, analyzed the data, prepared figures and/or tables, authored or reviewed drafts of the article, and approved the final draft.

The following information was supplied relating to ethical approvals (i.e., approving body and any reference numbers):

This study was approved by the Human Research Ethics Committee of Shanghai University of Sport (102772023RT008).

The following information was supplied regarding data availability:

The raw measurements are available in the Supplementary File.

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
