# Peer review of "Effects of 12-week integrative neuromuscular training on muscular fitness and sex differences in response to intervention in five- to six-year-old preschoolers"

_PeerJ, doi:10.7717/peerj.19417_

## Round 0.1 · original submission · Major Revisions

Dear Authors,

Please revise the manuscript considering the reviewers´ comments/suggestions.

Thank you.

Best regards.

**Language Note:** The review process has identified that the English language must be improved. PeerJ can provide language editing services - please contact us at [email protected] for pricing (be sure to provide your manuscript number and title). Alternatively, you should make your own arrangements to improve the language quality and provide details in your response letter. – PeerJ Staff

·

Basic reporting

The manuscript is well-written with clear and professional English. However, there are some minor grammatical inconsistencies and awkward phrasings that could be improved for better readability. A language edit by a native English speaker or a professional proofreading service would enhance clarity.

The introduction provides an adequate background and clearly justifies the study. The literature cited is relevant and up-to-date, covering the significance of neuromuscular training in preschool-aged children. However, there could be a more explicit mention of gaps in prior studies to further highlight the novelty of the work.

Figures and tables are well-structured, clear, and appropriately labeled. They effectively support the findings. However, Figure 4 could benefit from more descriptive captions to clarify the results.

Suggested Improvements:
Minor language and grammar revisions.
Enhance clarity in figure captions (especially Figure 4).
Strengthen the justification of the research gap.

Experimental design

The study is within the journal’s scope and addresses a relevant research question regarding the effects of integrative neuromuscular training (INT) in preschoolers. The study also explores sex differences, which is a valuable addition.

The experimental design is well-structured, following a randomized controlled trial (RCT) model with two groups (EG & CG) and three time points (T0, T6, T12). The methodology is appropriate, with sufficient detail to allow replication.

The study justifies its sample size using G*Power analysis, which is commendable. However, with only 30 participants (15 per group), the sample is relatively small. While the effect sizes appear large, a larger sample would strengthen the reliability of the findings.

The INT program is well-detailed, structured into progressive phases (adaptation, progression, and strengthening). Exercises are age-appropriate and follow evidence-based neuromuscular training principles.

Suggested Improvements:
Acknowledge the limitation of a small sample size in the discussion.
Ensure raw data files are properly annotated for full transparency.

Validity of the findings

The statistical approach is appropriate, utilizing independent samples T-tests and two-way repeated-measures ANOVA. However, it would be helpful to clarify whether assumptions for normality and sphericity were met before performing ANOVA.

Suggested Improvements:
Clarify normality and sphericity assumptions before ANOVA.
Acknowledge potential type II errors due to the small sample size in sex difference analysis.
Provide additional discussion on injury risk/safety measures in INT.

Additional comments

The study provides useful insights for kindergarten physical education. However, more discussion on real-world implementation (e.g., feasibility in large classrooms, teacher training) would add value.

·

Basic reporting

The introduction provides a strong background on the significance of muscular fitness in preschoolers and the role of integrative neuromuscular training (INT). Relevant literature is cited adequately. However, a clearer distinction between the contributions of this study and prior work would strengthen the novelty of the research.

The article follows a standard scientific structure (Introduction, Methods, Results, Discussion, Conclusion). Figures and tables are well-labeled and relevant to the study. Raw data has been supplied as per PeerJ’s policy.

The study is well-contained and directly addresses the research questions posed. The results align with the stated hypotheses.

Suggested Improvements:

1. Proofread for minor grammatical and syntactical errors.
2. Clarify the unique contributions of the study compared to existing literature.
3. Ensure figures are of high resolution for better readability.

Experimental design

The study aligns with the journal's scope as it examines an intervention relevant to exercise science and pediatric health. The research question is well-defined, focusing on the effects of a 12-week integrative neuromuscular training (INT) program on muscular fitness in preschoolers and exploring potential sex differences. However, greater emphasis on the specific knowledge gap being addressed would enhance the study's impact. Ethical approval is clearly stated, with informed consent obtained from parents, ensuring adherence to high ethical standards. The study design is robust, incorporating appropriate randomization and control measures. The methodology is detailed and provides sufficient information for replication, yet additional clarification is needed.

1. It is suggested to explain that there were no drop-out or deviations in both groups. This will help to assess the risk of bias to future researchers who systematically review the subject.
2. Indicate the conditions of clothing of individuals when measuring weight and height (barefoot, shorts, T-shirt?)
3. I suggest describing in more detail the activities carried out by the control group.
4. Describe the position in which the lower limbs were during the Core endurance strength test; were the feet free, were they held by a person, or on some device that allowed the kinetic chain to be closed?

Validity of the findings

The statistical analysis appears appropriate, with the use of independent t-tests and two-way repeated-measures ANOVA justified given the study design. Effect sizes are reported, enhancing the interpretation of the results. The conclusions are well-stated, accurately reflecting the results and remaining within the study’s scope, while the discussion effectively contextualizes the findings in relation to previous research.

1. In the discussion, the authors could emphasize more clearly the knowledge gap addressed by this research compared to previous studies.

Additional comments

I suggest improving Figure 3. Specifically, I suggest removing the background of the wall from each of the photographs since they visually lose some attractiveness. Ideally, the background would have been a wall of one color, without distractors such as windows. This comment is only to improve the quality of the manuscript.

I attaching an example of a study where the background of the exercises they show was removed.

https://www.mdpi.com/2411-5142/9/4/195

Reviewer 3 ·

Basic reporting

The manuscript is well-written in clear and professional English. The language is precise, and the flow of ideas is logical. Minor grammatical improvements could enhance readability. (see below)

Experimental design

The RCT design is appropriate for evaluating INT effectiveness. The randomization method (computer-generated) is well described. Some minor clarifications should be made to the manuscript. (see below)

Validity of the findings

Key findings are clearly stated. The EG showed significant improvements in all four muscular fitness measures compared to CG. Time × group interactions support the effectiveness of INT in improving muscular fitness. No significant time × sex interaction, suggesting both boys and girls benefited equally from INT. Additional information should be provided to clarify results. (see below)

Additional comments

1. Introduction
The introduction is well-written in clear and professional English. The language is precise, and the flow of ideas is logical. Minor grammatical improvements could enhance readability.
The introduction effectively sets the stage for the study by emphasizing the importance of muscular fitness in preschool-aged children. The research gap is clearly identified, and the introduction logically progresses to justify the need for the study.
One possible improvement is to add more context on why preschoolers are particularly underrepresented in integrative neuromuscular training (INT) research, beyond stating that research is limited.
It might be helpful to emphasize why gender differences in muscular fitness responses are expected at such an early developmental stage (to connect with the hypothesis).
The hypothesis is clearly articulated: both boys and girls will benefit from INT, but girls may be more sensitive to the intervention. However, the rationale for this hypothesis is based on prior research, but a more detailed justification focusing on physiological or neuromuscular reasons could strengthen this claim.
Minor language and clarity issues:

Lines 86–99: (paragraph discussing INT vs. traditional training)
This section is informative but somewhat dense. Consider breaking it into two paragraphs:
One focusing on how INT differs from traditional training methods
Another emphasizing its benefits and potential risks

Lines 126–129 (Hypothesis statement):
Adjust to a sentence more direct and eliminate passive construction.

Lines 121–125 (Sex differences in INT response):
It is stated that girls have shown greater improvements in some areas of INT, but there is no physiological or developmental explanation.
Consider briefly mentioning possible reasons such as neuromuscular adaptations, hormonal factors, differences in movement patterns or other relevant causes.


2. Methods
A few sentences are dense and could be rewritten for clarity. The explanation of statistical analysis and participant exclusion criteria.
The INT program details are thorough, but more information on adherence and dropout rates would be beneficial.
The participants were blinded to the study’s hypotheses, but researchers were aware of group allocations. Potential bias could arise from unblinded assessments, and this should be explicitly acknowledged in the limitations.
Were testers blinded to group allocation? If not, this could introduce bias in the assessment process.
The inclusion/exclusion criteria are comprehensive and justified. However, explaining why the three-month cutoff for prior INT participation was chosen could improve this chapter.
Statistical tests (t-tests or chi-square) should confirm whether groups were similar before intervention. Include this in table 1.
Explain how dietary and activity compliance was verified.
Explain why only two attempts were allowed for muscular fitness tests since most of the studies allow three trials for reliability.

3. Results
The results indicate no significant time vs sex interactions, yet sex differences in absolute improvements are reported. This contradiction should be clarified. If no significant interaction was found, absolute differences between boys and girls should not be overemphasized without further analysis.
Consider adding confidence intervals (CIs) for the mean differences to improve interpretability.
Were there dropouts or missing data? If so, briefly explain how they were handled.

4. Discussion
The lack of a significant time vs sex interaction means that boys and girls responded similarly to INT. However, the discussion emphasizes absolute percentage improvements in boys vs. girls, which may mislead readers into thinking the differences were statistically significant.
The sentence "This disparity could be attributed to their lower baseline performance than their counterparts, potentially enhancing their responsiveness to INT." should be supported with a statistical test if it’s a key claim. If not, this interpretation should be softened to avoid misleading conclusions.
The study compares its results with Faigenbaum et al. (2014), but differences in age groups and intervention protocols make direct comparisons difficult. The authors correctly acknowledge this, but further caution should be added when suggesting a "sex-specific developmental phase."
Additional study limitations should be considered, such as:
Blinding issues since researchers were not blinded to group allocation, which could introduce bias.
Adherence to INT (were all children able to complete the sessions or were there dropouts?). clarify previous chapters and adjust accordingly.
Potential confounding factors such as differences in daily activity levels outside of school.
Suggestions for improvement:
a. Break up long sentences for readability:
Original: "Meanwhile, preschoolers began to show significant improvements in muscular fitness at six weeks of INT."
Suggested: "Notably, significant improvements in muscular fitness were observed after just six weeks of INT. This suggests that even a relatively short intervention period can yield measurable benefits."
b. Soften overinterpretation of sex differences:
Instead of: "This disparity could be attributed to their lower baseline performance than their counterparts, potentially enhancing their responsiveness to INT."
Suggested: "Although boys and girls showed slightly different patterns of improvement, no significant time × sex interaction was found. This suggests that both sexes responded similarly to INT, despite some variations in absolute gains."
c. Explicitly state that sex-based trends were not statistically significant:
Add a clarifying sentence after reporting sex-based improvements, such as:
"While percentage increases suggest potential differences between boys and girls, these differences were not statistically significant."

5. Conclusion
The phrase "muscular fitness in preschoolers" appears twice in the same sentence.
The conclusion correctly highlights the practical applications of INT for physical education teachers and strength and conditioning coaches. However, a brief mention of policy or curriculum integration could strengthen the practical relevance.
The statement on long-term effects is good, but additional research questions could be posed, such as:
a. How does INT influence other aspects of physical fitness such as agility, endurance or coordination?
b. Are neuromuscular adaptations sustained after stopping the intervention?
c. How can INT be effectively integrated into the school curriculum?

---

## Round 0.2 · accepted · Accept

Dear Authors,

Thank you for your work during the review process.

Best regards.

·

Basic reporting

The authors have adequately addressed all the comments in the revised version of the manuscript. Therefore, I have no further comments.

Experimental design

The authors have adequately addressed all the comments in the revised version of the manuscript. Therefore, I have no further comments.

Validity of the findings

The authors have adequately addressed all the comments in the revised version of the manuscript. Therefore, I have no further comments.

Additional comments

The authors have adequately addressed all the comments in the revised version of the manuscript. Therefore, I have no further comments.

·

Basic reporting

The manuscript is written in clear and professional English, and the authors have stated that they sought both professional editing and native speaker proofreading. The flow of the text is logical and precise.

Relevant literature is adequately cited and provides sufficient context. The introduction presents a compelling rationale for the study, although the revised manuscript now more clearly highlights the knowledge gap regarding neuromuscular training (INT) in preschoolers.

Figures and tables are well structured, and improvements such as updated figure captions (e.g., Figure 4) and enhanced resolution contribute to clarity. Raw data has been shared and is appropriately labeled in supplementary materials.

The manuscript remains self-contained and provides results that directly address the stated hypotheses.

Experimental design

The methodology has been expanded and clarified per prior reviewer suggestions: sample characteristics, control group activities, measurement protocols (e.g., body position during tests), and statistical analysis assumptions are now more thoroughly explained.

The issue of small sample size has been acknowledged and addressed appropriately as a limitation, with implications for generalizability.

Ethical standards are well described, including informed consent.

Validity of the findings

Clarification regarding normality and sphericity assumptions has been added. Confidence intervals are now reported, which improves result interpretation.

The authors have appropriately tempered interpretations around sex differences by acknowledging the lack of statistically significant interactions, and they have clarified absolute improvements while avoiding overstatement.

Limitations are more explicitly stated, including small sample size, absence of blinding among assessors, and contextual challenges for generalizability.

The conclusions are well aligned with the research question and results.

Additional comments

The revised manuscript addresses prior reviewer comments thoroughly and professionally. Additions to the discussion on real-world implementation, teacher training, curriculum integration, and future directions (e.g., sustaining neuromuscular adaptations post-intervention) enrich the applied value of the study.

The revised visuals, methodological detail, and tone of interpretation have improved the manuscript substantially.

Reviewer 3 ·

Basic reporting

The manuscript is well-written in clear and professional English. The language is precise, and the flow of ideas is logical. Minor grammatical and syntax improvements recommended were made accordingly.

Experimental design

The RCT design is appropriate for evaluating INT effectiveness. The randomization method (computer-generated) is well described. Some minor clarifications were recommended to the authors. They reviewed it throughout the manuscript and edited it accordingly.

Validity of the findings

Key findings are clearly stated. The EG showed significant improvements in all four muscular fitness measures compared to CG. Time × group interactions support the effectiveness of INT in improving muscular fitness. Additional clarification in some key points was asked and the authors complied, resulting in an improved version of their manuscript.

Additional comments

None